# Analysis of Deactivation of 18,650 Lithium-Ion Cells in CaCl_2_, Tap Water and Demineralized Water for Different Insertion Times

**DOI:** 10.3390/s23083901

**Published:** 2023-04-11

**Authors:** Katharina Wöhrl, Yash Kotak, Christian Geisbauer, Sönke Barra, Gudrun Wilhelm, Gerhard Schneider, Hans-Georg Schweiger

**Affiliations:** 1Technische Hochschule Ingolstadt, CARISSMA Institute of Electric, Connected and Secure Mobility (C-ECOS), 85049 Ingolstadt, Germanysoenke.barra@thi.de (S.B.); hans-georg.schweiger@thi.de (H.-G.S.); 2EDAG Engineering GmbH, 85053 Ingolstadt, Germany; 3Hochschule Aalen, Materials Research Institute, 73430 Aalen, Germany; gudrun.wilhelm@hs-aalen.de (G.W.); gerhard.schneider@hs-aalen.de (G.S.)

**Keywords:** lithium-ion cells, cell deactivation, 18,650 lithium-ion cells, cell deactivation in tap water, cell deactivation in demineralized water, cell deactivation in CaCl_2_ solution

## Abstract

The deployment of battery-powered electric vehicles in the market has created a naturally increasing need for the safe deactivation and recycling of batteries. Various deactivating methods for lithium-ion cells include electrical discharging or deactivation with liquids. Such methods are also useful for cases where the cell tabs are not accessible. In the literature analyses, different deactivation media are used, but none include the use of calcium chloride (CaCl_2_) salt. As compared to other media, the major advantage of this salt is that it can capture the highly reactive and hazardous molecules of Hydrofluoric acid. To analyse the actual performance of this salt in terms of practicability and safety, this experimental research aims to compare it against regular Tap Water and Demineralized Water. This will be accomplished by performing nail penetration tests on deactivated cells and comparing their residual energy against each other. Moreover, these three different media and respective cells are analysed after deactivation, i.e., based on conductivity measurements, cell mass, flame photometry, fluoride content, computer tomography and pH value. It was found that the cells deactivated in the CaCl_2_ solution did not show any signs of Fluoride ions, whereas cells deactivated in TW showed the emergence of Fluoride ions in the 10th week of the insertion. However, with the addition of CaCl_2_ in TW, the deactivation process > 48 h for TW declines to 0.5–2 h, which could be an optimal solution for real-world situations where deactivating cells at a high pace is essential.

## 1. Introduction

Lithium-ion (Li-ion) cells are part of the modern technical and industrial world. They are widely used to store electrical energy for various mobile and stationary applications. The increase in demand has led to a rise in the production of Li-ion cells, their second use, and recycling. This recycling step after the second life helps to obtain a sustainable battery life cycle. Over-discharging a Lithium-ion battery (LIB) is usually necessary before initiating the actual recycling process. It can be carried out in different ways, such as the electrical discharging of the battery with a resistor or insertion into a conductive liquid [1].

This work focuses on the deactivation of LIB in different liquids (aqueous calcium chloride (CaCl_2_ solution), Tap Water (TW), and Demineralized water (DW)). This will help in the disposal of cells, modules, or the whole vehicle traction system, including deactivation in an extinguishing container where the cells are in critical condition [2]. Some of the major objectives of this research include looking into the deactivation process, i.e., the release of internal residual energy of a Li-ion cell, based on a variety of deactivation fluids, the deactivation process of CaCl_2_ solution in relation to TW and DW by considering the comparison of multiple different parameters such as conductivity, lithium concentration, pH value, fluoride concentration of the liquid, and cell Open Circuit Voltage (OCV).

From the literature, authors have identified that there are several different procedures for cell deactivation based on a variety of material compositions and concentrations of liquids [3,4,5], which makes it quite difficult to compare and contrast these procedures. An example of such research is the deactivation of a single cell by Lu et al. [3] and Li et al. [4] in the years 2013 and 2016, respectively. Both the researchers used aqueous sodium chloride (NaCl) solutions in varying concentrations; however, the outcome obtained by Lu et al. [3] was a rapid breakdown of the OCV of the cell within seven minutes after the cell was placed into a 10% NaCl solution, which indicates a fast discharge, while Li et al. [4], had a comparatively slower discharge, including the corrosion of cells phenomena. In the same research, further analysis of the metal ion concentrations of the liquid was also undertaken. Due to high concentration levels of several metals such as Sodium (Na^+^), Aluminium (Al^3+^), and Iron (Fe^2+^), it was concluded that all of these metal ions can be traced back to the casing of the 18,650 cylindrical cells, which means that the cell casing reacts with the liquid. Additional observation depicts a high content of phosphorous in the fluid, which, according to the research, results from the conductive salt Lithium Hexafluorophosphate (LiPF_6_) and indicates electrolyte leakage in the cell [4].

Shaw-Stewart et al. [5] inserted 18,650 batteries (fully charged) in 26 different liquid solutions for 24 h. The cell voltage, pH value, specific gravity, conductivity, and Electrochemical Impedance Spectroscopy (EIS) behaviour were tracked. It was found that the discharging process depends on the conductivity of the solution, the cell chemistry, and the competing reactions of the liquids. Furthermore, an electrolytic reaction due to Hydrogen gas was noticed at the cell’s negative terminal. This reaction was dependent on the pH level of the liquid and the corresponding reaction at the opposite electrode. On the positive cell terminal, water reduction took place, which resulted in the generation of gaseous oxygen. Shaw-Stewart et al. [5] also considered the research undertaken by Lu et al. [3] and interpreted that the OCV decrease was due to contact loss from the outside and found it improbable that the voltage loss represents actual chemical energy loss inside the cell.

Ojanen et al. [6] investigated the usage of varying aqueous solutions of salts like NaCl, NaSO_4_, FeSO_4_, and ZnSO_4_. In the experiment, the cylindrical cells were not inserted into the solutions, but only a set of platinum wires, which were fixed on both terminals on the cell, were inserted to measure the OCV. In this ex situ experiment, a 20% NaCl solution turned out to be an effective medium for the full and fast discharging of Li-ion cells within 4.4 h. Furthermore, it was also found that stirring the liquid can accelerate the discharging process, e.g., cells in the NaSO_4_ solution reached a deactivation time of 3.1 h. Moreover, it was also found that metallic particles are an effective way to reduce deactivation time and can prevent corrosion of the connectors. It is recommended to add these particles in a 20 wt% NaCl solution. The commonality between research [5,6] shows that the positive terminal of a LIB is mainly affected by corrosion, and consequently uncontactable terminal leads to difficulties during discharging, leading to an incomplete discharge.

Based on the above literature [3,4,5,6], it can be seen that the majority of the research focuses on the insertion of LIBs in different liquid media, conductivity of the liquids, comparison of the specific weight, and comparison of the impedance spectra before and after insertion. However, none of those authors have performed an experimental investigation on commercial 18,650 cylindrical cells, which are inserted into DW mixed with CaCl_2_. To the authors’ knowledge, until now, research related to deactivating the cells in mixtures with CaCl_2_ has not been considered, and hence this concept makes this experimental investigation as novel research. With all the aforementioned concepts, CaCl_2_ was chosen because of its availability, its nontoxicity, and its reactivity with the unwanted upcoming of the molecule HF.

Lithium Tetra Fluoroborate (LiBF_4_) and LiPF_6_ are the two commonly used salts in the electrolyte of LIB. Both salts possess the danger of Hydrofluoric acid (HF) formation when reacting with water or humid air. The chemical reaction sequence for the case of LiPF_6_ is demonstrated in Equations (1)–(3) [7],
LiPF_6_ → LiF + PF_5_(1)
PF_5_ +H_2_O → POF_3_ + 2 HF(2)
LiPF_6_ + H_2_O → LiF + POF_3_ + 2 HF(3)

However, CaCl_2_ with HF forms a poorly soluble CaF_2_ molecule along with HCL, as shown in Equation (4),
CaCl_2_ + 2 HF → CaF_2_ ↓ + 2 HCl(4)

In this research, the behaviour of CaCl_2_ solution is studied by inserting the cells into the water with soluble CaCl_2_ for different lengths of time. These results will also explain the cell deactivation behaviour and the state of safety over time. The additional goal of this research is to assess the time and respective critical points at which the cell deactivates, and the results will indicate the overall time period for which a cell needs to be inserted in a liquid media to be considered safely deactivated.

To research the efficiency of the deactivation of Li-ion cells with a CaCl_2_ solution, various comparative tests such as analysis of weight loss and cell insertion times into different media are carried out. DW and TW are used as reference deactivation mediums for the CaCl_2_ solution. Additionally, the comparison between CaCl_2_ and TW is made to recreate a more realistic review about deactivating the accident-damaged battery of electric vehicles in extinguishing containers [2] because TW is easily accessible and is widely used for fire/smoke extinguishing purposes.

Overall, this work aims to give a wide-ranging analysis of the effects of deactivation via different liquid insertions. To investigate the occurring damage to the cells after insertion, different measurements were carried out. For these experiments, each cell was given a specific identification number to make sure that they do not mix up during the destructive test, and these numbers are only used for reference during the destructive abuse test/analysis, i.e., serve the purpose of a tag (TW-24h-1, i.e., TW = Tap water; 24 h = Insertion duration of 24 h; 1 = Cell number 1).

## 2. Materials and Methods

### 2.1. Battery Samples and Characterisation

The selected test subjects are commercially available 18,650 Li-ion cylindrical cells (type: LiNiCoAlO_2_ (NCA); refer to Table 1 [8] for specification) in pristine condition. The reason for selecting the NCA cell is because this type of cell is suitable for a variety of applications, including electric vehicles, and is widely available commercially. After obtaining the cells from the vendor, they were all initially cycled ten times with a constant current (CC) charge and discharge at 1C during which the voltage limits from Umin to Umax (refer to Table 1) at room temperature were maintained to obtain a steady state. It is important to note here that the C-rate is being addressed here as per the nominal capacity of the cell given by the manufacturer. To cycle the cells (charging and discharging), a battery tester (Neware BTS 4008-5V6A, Neware Technology Co, Ltd., Shenzhen, China) was used [9]. After the completion of these initial cycles, the capacity and pulse resistance of all cells were checked at room temperature (19 °C ± 2.5 °C) to understand the deviating behaviour by comparing the capacity and resistance values. Furthermore, the results were additionally verified for the second time with a 1 kHz resistance measurement (HIOKI 3554 [10], Hioki E.E. Corporation, Nagano, Japan) before inserting them into a liquid medium.

### 2.2. Preparation of the Liquid Insertion Media

In this research, different insertion intervals are being investigated for different liquids. A distinction is made between the short-term (0.5 h to 10 h) and long-term tests (24 h to 3 weeks). For the short and long-term tests, a total amount of 0.4 dm^3^ and 0.8 dm^3^ liquid was used, respectively. As there is a real-time possibility that water may evaporate over a longer duration of time, and to make sure that all the cells are consistently underwater for the whole procedure, a higher volume of 0.8 dm^3^ was set up. Additionally, to avoid evaporation, the vessels were covered with plastic cling film throughout the experiment. Table 2 presents the distribution of cells in the different deactivation liquids. All insertion tests were carried out within laboratory conditions (19 °C ± 2.5 °C) in a fume hood. The time intervals were tracked manually with a watch to ensure that the tolerance was not more than a minute.

For the experiments with CaCl_2_, calcium chloride Dihydrate (CaCl_2_H_4_O_2_) with a purity level of ≥99% [11] was used and an aqueous solution of 2 mol/dm^3^ was prepared. Based on regulation VDE 0510-5, DW was used to dissolve the salt by thorough stirring. Scale (KERN ABJ-NM/ABS-N [12]) and a regular volumetric flask were used to prepare these solutions.

The temperature of the liquid during the test was monitored with a type K thermocouple at the centre of the vessel. The cells were inserted after the complete dissolution of CaCl_2_ at laboratory conditions. The deactivation medium was prepared and mixed for 24 h before each respective test.

To maintain consistency, the necessary amount of TW was filled in bottles from a regular tap at a single time before the experiment. The contents of the water are listed in Table 3. The TW had a water hardness of: 3.6 mmol/L CaCO_3_, 20.2 °dH, hard [13]. As per regulation VDE 0510-5, the reference tests were undertaken with commercially available chemically pure DW.

### 2.3. Procedure and Analysis Methods

#### 2.3.1. Test Procedure and Sample Distribution

Figure 1 illustrates the procedure of testing. Before inserting the cell in the prepared liquid, the cell was fully charged to Umax with a Constant Current, Constant Voltage (CCCV) step at 1 C, and with a cut-off current at 0.05 C. Thereafter, the cell was weighed with a scale, and the voltage, along with the AC resistance, was checked again to make sure that the values were within the standard operating range of an AC impedance measurement. While the deactivation process was ongoing in the respective medium, the liquid was continuously stirred [14] and the cell was fixed with a string so that the cell did not come in contact with the magnetic bar used for stirring purposes at the bottom of the vessel. During this procedure, each cell was placed in a separate chemical-resistant polyethylene vessel filled with the defined insertion liquid, as shown in Figure 2.

Table 4 demonstrates the sample distribution matrix for all deactivation intervals and analysis methods. Each symbol (o, x and ◊) within the time duration column presents the deactivation fluids (CaCl_2_, TW and DW, respectively) that were used for each analysis method. It is important to note that the long-term tests of 10.5 weeks for TW and DW were not initially planned, and therefore, the nail penetration analysis equipment was unavailable, which resulted in zero data. However, this does not negatively impact the overall results of this research, as this work is mainly focused on deactivating the cells in the CaCl_2_ solution, which occurred before the 10.5 weeks mark.

#### 2.3.2. Cell Removal Process

After each respective insertion interval (as shown in Table 4), the cell was removed (lifted out) from the medium with the help of an attached string and dried with paper cloths. On both terminals of the cell, crystal-like structures were formed due to the deactivation process. To access both terminals for the voltage measurement, the incrustations, as visible in Figure 3, were removed. Figure 3 is an example of a cell inserted into TW for two weeks. After that, the liquid was filtered through a chemical filter (CONATEX 200 mm [15], Conatex-Didactic Lehrmittel GmbH, Saarbrücken, Germany) into a chemical-resistant polyethylene bottle to filter out coarse particles (Figure 2, bottom left and right). This sample was used for chemical analysis in a later step. After 15 min of the rest phase, the voltage and the mass of the cell were checked again. Positive and negative terminals of the cell were sealed with Parafilm to limit the evaporation or leakage of the cell as much as possible. Thereafter, the cell was stored to transport for the nail penetration test at a designated area.

#### 2.3.3. Nail Penetration Test

Based on the authors experience, to investigate the remaining internal chemical energy of the cells, a destructive test needed to be performed, i.e., inserting a steel nail into the cell. To yield comparable results, care was taken to ensure that the time interval between the removal of the cell from the liquid and the penetration of the nail was between 2 h and 3 h for all tests. Figure 4 shows the self-made nail penetration apparatus from Technische Hochschule Ingolstadt, Germany. The steel nail (thickness of 3.5 mm and length of 65 mm) was mounted in a plastic thread, which was screwed into the pneumatic piston, as shown in Figure 4 on the bottom right picture. The tip of each nail was manually bevelled and flattened—similar to a screwdriver head (Figure 4, top right)—to make sure that it penetrated the jelly roll of the cell as well as reached the active material. The nail was controlled pneumatically via compressed air (5 bar) using a compressor and valve regulation. The cell was fixed on a small self-constructed mounting table and tightened to a metal wall so that it did not slip away when the nail hit the cell. The K-type thermocouple (range up to 1200 °C) was fixed with an elastic material (Figure 4, bottom right) to the middle of the cell, which was monitored and recorded during the experiment. Before starting the test, the cell was aligned correctly to make sure that it hit the centre of the positive terminal of the cell. The nail penetration was allowed to reach the depth of 45 mm into the cell to check if the cell contained enough residual internal energy, and if so, it would trigger the exothermal reaction, which would lead to fire due to the internal short circuit caused by the nail. For such cases, a CO_2_ fire extinguisher was used to prevent damaging the nearby measurement equipment.

#### 2.3.4. Chemical Analysis of the Liquid after Test

After each insertion period and subsequent removal of the cell, the liquid was filtered and transferred to a Polypropylene bottle (see Figure 2 bottom left and right). After filtration, the complete liquid was stirred and 10 cm^3^ of liquid samples were taken separately for each analysis method and for each deactivation fluid, respectively. The investigation of this sample was carried out to understand the status of the cell opening and its decomposition. For this purpose, four different analysis methods were used—(i) Flame photometry, (ii) Fluoride electrode, (iii) Conductivity, (iv) pH value—with all investigations conducted in laboratory conditions (19 °C ± 2.5 °C).

As the focus of this work was to determine the Li-ions in the sample that indicates the cell openings and liquid exchanges from the inner parts of the cell, the digital flame photometer (FP910, pg instruments [16]), capable of determining the contents of Lithium, Sodium, Potassium, Calcium, and Barium, was used in the experiment. In total, 25 samples of liquids were tested simultaneously in a single run. The photometer was calibrated with a calibration solution at 0 ppm, 5 ppm, and 30 ppm. Between the deactivation liquid analysis of each 10 cm^3^ sample, the suction tube of the Flame Photometer was cleaned with DW to prevent sample-to-sample contamination.

To detect fluoride ions in the samples, an ion-selective electrode (F 60, SI Analytics [17]) was used in combination with a TISAB (Total Ionic Strength and Buffer) solution (WTW, Xylem Analytics Germany GmbH, Weilheim, Germany). The detailed measurement procedure is described in [18]. Two-point calibration of the ion-selective electrode was used with a 0.1 mol/dm^3^ and 10 mol/dm^3^ fluoride standard solution. The lower calibration point could not be set any lower due to device limitation, and the upper calibration point was set to 10 mol/dm^3^ as the real sample data was expected to be in the lower range, i.e., not more than 10 mol/dm^3^. Measured values below 0.1 ppm concentration were not considered for analyses, as this range lies below the calibrated minimum value of the ion-selective electrode. For the sampling, 2 cm^3^ of the TISAB solution and 2 cm^3^ of the filtered and prepared deactivation liquid were mixed and analysed by the ion-selective electrode.

The conductivity of each insertion media was determined by using a conductivity meter (GLF 100, Greisinger, Regenstauf, Germany [19]). This device had a measuring range from 100 mS/cm to 2000 µS/cm. Due to the high conductivity of the CaCl_2_ samples, the conductivity meter was not able to determine the actual value, and thus, a 1:5 dilution of those sample probes was used. In total, 10 cm^3^ of the filtered sample probe was filled in a 50 cm^3^ volumetric flask and mixed with four parts of DW. The conductivity of DW and TW was in the measuring range of the conductivity meter, so no dilution was necessary for those cases.

Furthermore, the pH value of each liquid was determined with an electronic pH measuring device (GMH 5550, Greisinger, Regenstauf, Germany [20]). The measuring probe was inserted into the 10 cm^3^ filtered solution of each sample. During the measurements, a constant ambient temperature of 21 °C was maintained.

#### 2.3.5. Computer Tomography (CT)

CT was used for the non-destructive investigation of the whole battery. This allows for a macroscopic characterisation of changes inside the cell due to the deactivation reaction. The studies were performed using a Phoenix V/tome/X CT (microfocus tube) with a voltage of 120 kV and a current of 100 μA. The exposure time was 200 ms and the maximum resolution was 22 μm voxel size.

As the focus of this work lies on the deactivation of Li-ion cells via CaCl_2_, cells inserted in the liquid containing CaCl_2_ were additionally analysed by CT measurements at Hochschule Aalen. The manufacturing defects were analysed by measuring the cell before the investigation. Table 4 represents the intervals at which designated cells were analysed for their changes in structure over time. In addition to the intervals at ten hours, forty-eight hours, and one week, an extra investigation at five minutes insertion was considered, as the positive terminal showed rapid reactive behaviour when inserted into CaCl_2_ solution. For reference, a single cell in pristine condition was scanned to visualize changes happening due to the experiment.

## 3. Results

To structure the experimental results in this section, three categories are considered. The first category is described in Section 3.1, in regard to the cell voltage and weight before and after the nail insertion, followed by the results of the nail penetration tests (Section 3.2) and the chemical analysis of the deactivation liquid (Section 3.3).

### 3.1. Voltages and Weights of the Cells

As described in Section 2, the cell voltage and cell weight were recorded 15 min after the removal of the cell from the deactivation liquid. The measured cell voltage for each deactivation period is shown in Figure 5. The initial OCV before insertion was U_init_ = (4.18 ± 0.01) V for each cell and is represented with a black dashed line in Figure 5. The colours of the bars represent three different insertion fluids. The height of each bar portrays the mean value of the measured voltage. Furthermore, each bar is equipped with a corresponding tolerance band which represents a standard deviation of one Sigma (σ). For the behaviour of the cells deactivated into CaCl_2_ solution, it was noticed that there was a strong monotone decrease until the 10 h readings. The longer insertion intervals led to a similar behaviour, with a final voltage circulating around 0 V and even crossing the negative range for 5 h, 10 h, 24 h, 48 h, and 2 weeks. From 0 h to 0.5 h, a negligible voltage drop was observed, i.e., the divergence of voltage loss was only 0.09 V, whereas the readings between 2 h and 5 h showed a significant voltage loss from 2.7 V to 0.27 V, respectively.

In contrast to CaCl_2_ solution readings, the cell voltages for TW and DW levels were close to each other at 4.15 V till 24 h. Starting from 24 h, the difference between both groups begins to increase as the voltage decrease of TW cells accelerates and ends at 0.11 V at 10.5 weeks. After 2 weeks and 10.5 weeks of insertion in TW, the voltage readings were 0.85 V and 0.11 V, respectively. Comparing the cell inserted into TW with DW, it can be seen from Figure 5 that the DW cells showed almost no voltage drop till 2 weeks of insertion, i.e., the value was at 4.15 V. Though, from 2 weeks to 10.5 weeks, the voltage dropped to 2.35 V. Based on these measurements, it can be said that inserting the cell in CaCl_2_ solution results in a faster voltage loss compared to TW and DW, i.e., cells inserted in CaCl_2_ solution require around only 5 h for cell deactivation, while the deactivation process with DW took more than 2 weeks to start.

Before insertion, each cell was weighed, and all measured values led to an overall average of 43.18 g. This initial mass m_Init_ is depicted in Figure 6 as a black dashed line. Like the colour scheme of Figure 5, the insertion fluids CaCl_2_, TW, and DW have been assigned the same corresponding colours. The insertion intervals are plotted on the x-axis, and the mean of the measured cell mass after the removal of the liquid at each respective condition is plotted on the y-axis. In addition, each bar is equipped with its one σ standard deviation, giving an overview of the distribution of all three measurements at that condition, except for the 0 h condition where 19 readings of CaCl_2_ solution and 18 readings of TW and DW each are averaged.

The cells deactivated in CaCl_2_ solution had a declining mass up to 42.83 g (equivalent to 0.8% when compared against m_Init_) from the beginning of 5 h results, and thereafter it inclined up to 43.7 g in the 2nd week. After the removal of the cell from the liquid, deposits on the cell terminals such as salt incrustations were observed. However, all cells were carefully dried and cleaned (off-incrustations) before weighing.

The phenomenon of weight increase was also observed for the samples of TW and DW. Especially for the long-term tests (≥1 week), there is an increase of up to 44.4 g for 10.5 weeks insertion in TW, which translates to a relative increase of 2.83%. The newly formed incrustations were again removed from these samples before weighing.

According to the authors, the phenomena of increment in the weight is because the liquid enters the cells (especially during long-term insertion), and the inner parts of the cells absorb the liquid and react with it internally in a similar manner as on the outside of the cell terminal (incrustations, Figure 3). This situation cannot be avoided, and therefore, the cell has a weight increment. The intrusion of material or liquid can also be seen in the CT analysis later.

### 3.2. Nail Penetration Test

The assessment of the internal residual energy of all deactivated cells is quite a challenging step, and therefore, the energy was determined qualitatively by considering the cell behaviour during abusive conditions. Such results are presented in this section with the help of Figure 7, Figure 8, Figure 9, Figure 10, Figure 11, Figure 12, Figure 13, Figure 14, Figure 15 and Figure 16. For each deactivation fluid section, two samples are tested. In these graphs, indices are given to each measured cell where the first part of the label equates to the insertion duration and the second part to the index of the cell, i.e., Ca 2h-1 being the first sample deactivated in CaCl_2_ for 2 h. These figures depict the temperature curves of the cells as the nail hits the centre of the positive cell terminal. The time of nail impact is presented with a vertical red dashed line.

Figure 7 shows all measured temperatures for 0.5 h deactivated cells in different deactivation media. Cells that were inserted in the CaCl_2_, TW, and DW solution are depicted in blue, red, and green, respectively. The initial temperature values before the penetration of the nail were between 35 °C and 60 °C. The reason for such a large range is that one of the sensors caught fire due to an unknown reason, and henceforth, the sensor had to be changed. All tested cells in Figure 7 are burning as a result of the nail hitting the cell. The temperatures remained in the range of 250 °C to 430 °C. The behaviour of all cells independent of the deactivation media proves that there is significant residual energy in the cells to ignite. Therefore, it can be inferred that an insertion time of 0.5 h is not enough to deactivate the cells.

Figure 8 presents the comparative results of the cell reaction after the cell was inserted for 2 h in the CaCl_2_ solution, TW, and DW. After nail penetration, the temperatures of the TW and DW raised to 483 °C and eventually resulted in a fire. This can be seen in Figure 9 (bottom left and right). Whereas the cells that were deactivated in CaCl_2_ showed neither temperature increase nor smoke/fire (Figure 9, top left and right), meaning their residual energy was drained enough to avoid ignition.

As in Figure 8, the nail penetration results from the 5 h deactivated cells are presented in Figure 10. It is evident that all three cells tested in the CaCl_2_ solution showed neither a drastic temperature increase nor smoke/fire. This confirms the outcome from the 2 h results (refer to Figure 8 and Figure 9). On the other hand, the samples of TW and DW showed rising temperatures trend till 430 °C and 420 °C, respectively, along with some flames. All four samples of TW and DW had some sparks and flames and eventually completely ignited (on fire).

Figure 11 illustrates the temperature curves after 10 h insertion in different media. As in Figure 8 and Figure 10, cells inserted in CaCl_2_ solution showed neither temperature change nor smoke/fire. One of the two samples, which were deactivated in TW, still led to temperature increase and fire, whereas the second sample had neither a significant temperature rise nor fire.

In Figure 12, the temperature curves of 24 h inserted cells are presented. One sample cell that had been inserted into CaCl_2_ solution showed rising temperatures of about 36 °C, and the video recordings of this cell also showed some development of smoke, whereas no other sample of this category showed any reaction. Similar to Figure 10 only one sample of TW was ignited, while the other ones displayed no reaction and no temperature rise. However, both samples of DW indicated enough residual energy inside to ignite.

Figure 13 shows the temperature curves of 48 h cell insertion results in CaCl_2_ solution, TW, and DW. None of the CaCl_2_ solution inserted cells showed fire occurrence, but one of them had rising temperatures from 32 °C to 72.3 °C along with the smoke development, i.e., which means there was some residual energy left in the cell to ignite. The samples of TW and DW showed a strong increase in temperature which led to ignition, sparks, and fire.

The results after 1 week of insertion for cells in TW and DW are shown in Figure 14. The prominent noticeable result at this stage is that both the samples of TW had no smoke/fire. Only one TW cell (TW-1W-1) displayed an increasing temperature of up to 91.7 °C, while both the DW samples reacted and had an increased temperature, which resulted in a fire.

The temperatures during the experiment for the 2 week insertion period are displayed in Figure 15. There was no reaction at all for both samples that were inserted into the CaCl_2_ solution, as well as for the samples in TW. However, the temperature of one cell deactivated in DW (DW 2W-1) rose to 79.6 °C with the development of little smoke. Consequently, it can be said that this cell and both other test groups did not have enough residual energy for a pronounced exothermal reaction. It is only the second DW cell that reacted heavily (DW 2W-2) with rapidly rising temperatures and fire.

Altogether, it can be noticed that the cells deactivated in a CaCl_2_ solution led to a reduced thermal reactivity, with no exothermal reaction present between the interval of 0.5 h and 2 h. Before this time interval, all cells showed a comparatively similar temperature increase. After 5 h insertion time, one of the TW cells showed decreased reactivity, reaching approximately 200 °C, and after 10 h of duration, no reaction was identified. After 24 h, one DW cell also showed reduced exothermal reactivity. After 2 weeks, one of the DW cells showed a reduced reaction, while the second DW cell (DW 2W-2) was still reacting and reaching its peak of 322 °C.

Figure 16 shows the exothermic reactions based on the EUCAR hazard rating level. For example, at 2 h insertion, the rating dropped from level 6 (meaning occurrence of fire and flying parts) to level 2 (which means no fire occurrence but a cell defect). For TW cells, there was no clear pattern, i.e., between 2 h and 48 h of cell insertion, the hazard levels were fluctuated significantly and reached up to level 6 at 48 h hours of insertion. This means that, from a safety point of view, TW deactivation is not as reliable as using the CaCl_2_ solution. Cells deactivated in DW showed high reactive behaviour up until 2 weeks of insertion, depicting even worse deactivation behaviour than TW.

### 3.3. Analysis of Liquid

#### 3.3.1. Flame Photometer

With help of flame photometry, the three distinct liquids were probed for the occurrence and concentration of Li-ions (refer to Figure 17). It is important to note that the CaCl_2_ solution was only 99.9% pure, and hence, it had other potential compounds including Li-ions. For the cells inserted in TW, the values were in a nominal range, and from the 1st week, the Li-ions concentration increased slowly and reached 0.45 ppm after 10.5 weeks of insertion. For DW, the detected Li-ions concentration was constantly below the lower detection range until 48 h. Thereafter, from the 1st week onwards and until 10.5 weeks, Li-ions were recorded, but only at the lower detectable limit of around 0.1 ppm.

#### 3.3.2. Ion-Selective Electrode

The experiment conducted at this stage was also similar to previous experiments because it had three distinct measurements for CaCl_2_ solution inserted cells and two for TW and DW each, which were recorded and averaged (see Table 5). As the measurement range was physically limited to 0.1 ppm, readings below this limit are not meaningful and should be discarded. Though, for reference purposes, Table 5 presents all the results. It can be noticed from the table that the value of the CaCl_2_ liquid after 10 h and 48 h are close to the limit of the minimum resolution range of 0.1 ppm fluoride content, and therefore, it can be said that no significant amount of fluoride is present in the CaCl_2_ samples.

The recorded concentration values of TW show reduced significance in the first eight samples and a clear rise after 10.5 weeks (up to 11.3 ppm fluoride content). Before that, all values are relatively similar and below the 0.1 ppm detection range, except for the 1st week reading, which is 0.105 ppm. As all values before 10.5 weeks lie in a similar value range close to the lower detection limit, the authors interpret this as an unopened cell with the opening taking place between the 2nd week and 10.5 weeks.

The readings of the DW samples show reduced significance, as all measured values are below the tolerance limit of the equipment, but also provide the information that there are no free fluoride ions in the liquid, and thus, no opening of the cell.

#### 3.3.3. Conductivity

Figure 18 shows the measured conductivity in mS/cm of the insertion fluid after filtration. The x-axis displays the duration of the insertion of the cells. From the recorded readings, it can be noticed that the graph bars with CaCl_2_ solution (in blue colour) reach the highest values of measured conductivity and are—even with a dilution of 1:5—in the range of 55 to 57 mS/cm, while at 2 weeks intervals the value reaches up to an average of 60 mS/cm.

The conductivity of TW is measured with an undiluted sample (red colour bars), and all values of each insertion category are within the range of around 0.5 mS/cm. On the other side, the measuring bars of DW (in green colour)—also not diluted—can be found at around 0.01 mS/cm.

#### 3.3.4. pH Values

Figure 19 represents the average of the measured pH values of the deactivation media after the insertion and removal of the cells. The dashed black line marks the pH neutral value of 7. The blue bars indicate that the pH average of the CaCl_2_ solution starts at 6.33 and reduces over time. For example, the long-term test of 10.5 weeks shows a pH value of 5.16, consequently the solutions are lightly acidic. The decrease in pH value is due to the formation of HCl, as explained in Equation (4) in Section 1. Considering the TW value, it can be noticed that the pH values are between 7.8 and 9 for the 0.5 h to 48 h tests, respectively. Therefore, it can be said that the pH values are in the lightly alkaline category. However, the long-term tests (≥1 week) indicate decreasing pH values, i.e., between 7.1 to 6.7, and henceforth can be categorized into acidic pH values. For DW, all the samples were alkaline, though nominally aligned towards neutral pH values at a longer deactivation interval.

### 3.4. Computer Tomography (CT)

Figure 20 presents the results of the CT scan of the cells. In the top row (a–e) of Figure 20, the cell is viewed from an aerial perspective, with the jelly roll in the centre and the plus pole on top, whereas pictures (f–j) show the side view of the cell. It should be noted that the series of pictures (a–g) are not from one cell but from different cells investigated over time.

From Figure 20, pictures (a,f) portray the initial state before deactivation. A thin metal foil starts between the active material layers and contacts the cell coil to the positive pole. The characteristic visible gap forms below this contact. After five minutes cells being inserted in CaCl_2_ solution, a slight deformation of the upper jelly roll is detectable (see Figure 20b). This phenomenon is further expressed at 48 h of insertion, as can be seen in the scans (refer to Figure 20d,i). In the side view of Figure 20h,i,j it can be seen that the structures form internally between the cell layers, leading to the displacement of the sheets and to dissolution processes of the active material. The deformation in the middle of the jelly roll is mostly expressed after 1 week in (Figure 20e). The greatest damage in the side view was observed for the cell stored in CaCl_2_ for 48 h (Figure 20i). The authors assume, that the deformation of the jelly roll facilitates the penetration of CaCl_2_ leading to faster dissolution of the active material.

## 4. Discussion

Based on the distinct insertion media (CaCl_2_ solution, TW, and DW) used in this investigation for cell insertion, this section provides a discussion about the results obtained from the experiments.

### 4.1. Insertion in CaCl_2_ Solution

Considering the OCV curve of the cells, after the deactivation of cells in CaCl_2_ solution, a voltage drops to 0.27 V after 5h can be observed as an over-discharge phase. After 2 h of insertion, a voltage drop close to the Umin (0% SoC) of the cell was determined. Theoretically, the voltage loss can derive from two causes: (a) the electrical energy of the cell could be discharged until it reaches the referenced voltage level, or (b) the terminals lost contact with the electrodes over time due to the corrosive reactions with the fluid. However, the authors suspect this voltage decrease can be attributed to the actual loss of electric energy of the cell, as this aligns with the results of the residual energy determination of the destructive test. The biggest voltage change (from more than 4 V to 0.25 V) occurred between 0.5 h and 5 h, and before this interval, the positive terminal reacts with the fluid and it dissolves (see Figure 5). This is also supported by the measurement of the cell mass, as after 0.5 h, a slight decrease in cell mass (0.6%) was noticed, which can be attributed to the reaction of the terminals with the fluid (dissolution). At the 5 h insertion period, an increase of 2% in weight was noted, while the maximum weight (approx. 43.5 g) was noticed during the 2nd week (refer to Figure 6). The authors attribute this to the opening of the cell and the reaction of the fluid with the inner materials, where incrustations may have formed over time and have contributed to the weight gain. This also aligns with the results from the CT scans in Figure 20 at 48 h and the 1st week.

No drastic temperature rises or fires occurred during the destructive test conducted after 0.5 h (refer to Figure 7). Even though the voltage of the 2 h deactivation cell was still above its minimal voltage according to the datasheet, no thermal reaction of the cell took place during the abusive test, which indicates only a limited amount of residual energy. This supports the deduction from the voltage analysis.

For the flame photometer, the initial lithium values were in the same range as subsequent values, leaving no significant change. From the measurement of the ion-selective electrode, no significant readings for any fluoride release were detected. All recorded F- values were below or nearby the minimum detection limit of 0.1 ppm, and thus cell opening time was unable to be analysed from the experiment.

For the conductivity reading of the CaCl_2_ fluid, an increment of 8.1% (up to 60 mS/cm) was noticed between 10 h and 2 weeks. It was expected that the release of additional ions from the inner parts of the cell would not lead to a drastic increase in conductivity, as the electrical resistance of the liquid was mainly dominated by the dissolved salt CaCl_2_ (2 mol/dm3 concentration). Although, such a high increase (8.1%) could be because of the cell being ruptured between 10 h and 2 weeks.

When considering the pH value of the CaCl_2_ solution, all values were within the acidic category. The analysis of the pH value over time showed a declining trend from 0.5 h to 2 weeks. Though, the maximum fall in value was noticed between 48 h and 2 weeks, i.e., from pH 6 to pH 5.16 (refer to Figure 19).

### 4.2. Insertion in TW

Looking at the OCV curve for cells that were deactivated in TW for 1 week, it can be said that the cells had a medium range voltage with 3.56 V. Furthermore, the value dropped at 0.86 V and 0.11 V for 2 weeks and 10.5 weeks, respectively, for which it can be considered as an over-discharged cell (based on OCV curve). For the CaCl_2_ solution, the voltage loss can be assigned to the two aforementioned phenomena: loss of contact with the electrodes or electrical discharge. From visual observations of the cell by the authors, both terminals—especially compared to the positive one in CaCl_2_—were still in a close to pristine condition. This voltage loss is attributed to the electrical discharge over the ions of the TW. This hypothesis can be supported by the conductivity readings, which were not as high as the CaCl_2_ solutions, but were still not as low as the value from DW. Additionally, the voltage loss between 48 h and the 1st week indicates a cell opening scenario. The temperatures of both test cells during the 48 h nail penetration for TW reach a maximum at 373.4 °C and 476.5 °C, respectively, with a steep incline in the beginning period, and thereafter, at 1 week insertion, the temperature curve for one cell was completely flat at 37.3 °C and the second cell showed minimal exothermal behaviour up to 91.8 °C. Both cells did not show any signs of fire, and thus, the residual stored energy must be very low. In particular, one cell after 10 h insertion also showed negligible exothermal behaviour, even though the longer insertion intervals (24 h and 48 h) do react in an exothermal manner. The authors credit this phenomenon to a probable cause in which the nail would be because of nail bending or the nail not hitting the active material of the cell correctly during the test.

During the period from 48 h to the 1st week, the most remarkable change in weight occurs, i.e., from 43.5 g to 44.1 g (1.4% increment; refer to Figure 6). This can be attributed to a cell opening and internal reactions in the cell as well as to some residual remains from the liquid inside of the cell because the outer parts have been dried with a paper cloth and additionally air-dried for 15 min. It is also noteworthy that no initial weight decrease was observed as the terminals did not dissolve in reaction with the fluid compared to the CaCl_2_ solution situation.

Regarding the readings of the flame photometer, the initial Li-ion values from 0 h to 48 h were in the range of 0.1 ppm. Thereafter, the Lithium concentration raised slightly up to 0.15 ppm between the 1st and 2nd week and reached a peak of 0.45 ppm at 10.5 weeks, which indicates a cell opening between 48 h and by the end of the 1st week.

The recorded F- concentration of TW showed reduced significance up to 48 h. For example, in the 1st week, one of the cells had a concentration of 0.046 ppm, and the second one had 0.163 ppm. After 2 weeks, this value reduced to 0.026 ppm and 0.089 ppm, and thereafter, it increased to 13 ppm and 9.67 ppm at 10.5 weeks. Between 48 h and the end of the 1st week, the fluoride content for one cell was above the detection range, which could be due to a cell opening for at least one of the cells and a reduction in residual energy. This matches with the behaviour of both cells during the nail penetration test in the 1st week, where one cell showed exothermal behaviour. More specifically, cell 81, which has a 0.046 ppm F-concentration and thus was less likely to have opened in the 1st week, showed higher exothermal reactivity during the abusive test (see Figure 14), whereas cell 119, which has a 0.163 ppm F-concentration, implying to a cell rupture, shows the flat thermal behaviour at the nail test. For the 2 weeks experiments, no exothermal behaviour was detected, so there were no F-concentration readings. The authors suspect the reason for this contradictive behaviour to be F-concentration readings close to the detectable range of the device. This, in addition to the remaining unchanged Li-ion concentration from the flame photometry analysis for 1 and 2 weeks, suggests that the cells at 2 weeks should already have opened and decreased in residual energy. After 10.5 weeks, the cells were ruptured and reacted with the solution.

The conductivity readings of the TW displayed a slow monotonic decrease from 0.65 mS/cm at 0.5 h to 0.46 mS/cm at 48 h. From there on until the end of the 1st week, the reading dropped to 0.3 mS/cm. This 34% decline can be attributed to changes within the cell, i.e., it is suspected that ions within the TW may have reacted with the inner parts of the cell, and thus, they are unavailable for charge transport, which results in reduced conductivity. A similar pattern was noticed for the pH analysis as well. The initial readings up to 48 h were in the alkaline category, i.e., close to pH 8. At 48 h it reached pH 9 and dropped to the neutral area at pH 7.1 by end of the 1st week, moving further into the acidic domain. Fitting to the scheme, the most apparent change happens between 48 h and the end of the 1st week, i.e., change occurs due to the reaction of H_2_O and LiPF_6_ creating HF.

### 4.3. Insertion in DW

The OCV curve readings after the deactivation in DW showed no significant variation for up to 2 weeks. The last recorded value at 10.5 weeks shows the average voltage drop to 2.35 V and only one cell being at 4.08 V. This behaviour can be explained by the low conductivity of DW, as can be seen from the conductivity measurements. The low conductivity leads to the slow self-discharge of the cell in DW. From 0 h to 0.5 h the reading was very low, and, according to the authors, this could be due to some small particles of dust or salt possibly depositing on cells while handling them for the experiment. Overall, the conductivity measurement shows a 48% increase over time, from 0.5 h with 13.5 µS/cm to 20 µS/cm at 10.5 weeks.

The cell mass was nearly constant (at approx. 43.2 g with ±0.2 g) until 48 h of insertion, but then increased by 2.2% at the 1st week and stayed at that level until 10.5 weeks (refer to Figure 6). This shift can be interpreted due to potential cell openings like TW and CaCl_2_ solution. Considering the temperatures during the destructive nail test for the 2 weeks insertion, a noticeable observation for TW cells can be made. One cell (DW 76) reacted in a similar exothermal manner as the cells with lower insertion times, with a peak temperature of 322.4 °C, whilst the second cell (DW 84) showed a flattened curve, only reaching 79.6 °C without fire and ignition. From this, it can be said that, for deactivation in DW, the reactions that lower the residual energy content of the cell take place between the 1st and 2nd week, even though no distinct voltage loss is visible from the outside. A possible explanation of such could be due to the high impedance behaviour of the cell that did not react exothermally. From the destructive test for the DW cell at 2 weeks, it can be said that at least some energy was lost before, with one cell staying below 80 °C and without fire. This means that even though a high voltage is still present at 2 weeks, a reduced thermal reaction is observed, leading to the outcome that the voltage may not be a reliable indicator for all situations regarding deactivation.

When considering the readings of the Lithium concentration over time, no Lithium was found for the period of 0.5 h to 48 h. From the 1st week onwards, a Lithium concentration of 0.05 ppm started emerging and reached up to 0.1 ppm till 10.5 weeks. As DW is considered pure water (compared to 99.9% of CaCl_2_), no impurities from the solution are to be expected passed on. However, if the impurities are coming from the equipment, it is improbable that three distinct measurements contained the same error. Thus, the detected Li-ions are suspected to originate from the interior of the cell.

The readings of the Fluoride concentration of the DW samples showed quite low-value readings, i.e., below the tolerance of the equipment, and therefore, no significant additional information was gained. Similarly, the pH progression over time was comparatively stagnant between pH 8 and 9 for the interval of 0.5 h to 48 h. Thereafter, a negligible trend was identified towards neutral, and therefore, it can be said that the overall trend for the complete data set is alkaline.

### 4.4. Predictive Capabilities of the Measured Parameters

Throughout the experiment, various parameters were recorded, but not all readings were suitable for the determination of an accurate state of a cell during deactivation in a liquid medium. Therefore, this topic is addressed in this section, starting with the parameter voltage. The behaviour of the DW cells after 2 weeks of insertion was so noticeable that, even though a nominal voltage was close to full SoC before the destructive test, both cells reacted with reduced exothermal behaviour during the nail penetration test. The other reason towards to voltage parameter not being a completely reliable parameter is because of the dynamic reaction of the outer cell terminals with the fluid, especially in the case of the CaCl_2_ solution. The risk that emerges here is the faulty voltage readings, i.e., a false result due to the loose contact between the terminal and the electrode and due to incrustations at the terminals.

Another important parameter is cell mass. This parameter can be used for evaluating processes such as the dissolution of the positive terminal or formation of incrustations. However, in real-world applications, this is not pragmatic when considering larger battery packs. The additional way that can be used for identifying the activation/deactivation status of a cell is to undertake the mechanical destructive test (abusive test). The issue with this approach is that this method can be used for analytical investigations but not for analysing the large battery packs from the wrecked EV. The reason for this is that, in many cases, it is impossible to remove the cell from the pack.

One of the other approaches is to look for the lithium concentration in the fluid and identify whether the cell casing has opened and if there are any major impurities in the solution. However, during the experiments, it was found that the value of impurities was below the detection tolerance of the measurement setup, and therefore, no readings were identified, except for the last insertion period for DW, where nominal readings were recorded. Therefore, with the help of this approach, it can be said that, if there is a deflection in reading, then the cell is deactivated.

Moreover, the conductivity parameter could also be used for evaluation purposes, but the issue with such a parameter is that the influence of other ions present in the solution, along with the potential dirt and dust, can falsify the results. Similarly, the pH value as a parameter is also not well-suited as a small change in the concentration of H30+ or OH- ions may lead to a high change in the pH value, i.e., demonstrating drastically wrong results.

## 5. Conclusions

In this research, various aspects (mass loss, voltage loss, pH, conductivity, Lithium and fluoride concentration, and abusive behaviour) of three different media (TW, DW, and CaCl_2_ solution) were analysed and compared against each other to see if they demonstrated a safe and reliable deactivation process of Li-ion cylindrical cells. An exemplary general overview of the relative changes in the various parameters is summarized in Figure 21.

From Figure 21, it can be seen that the addition of the salt CaCl_2_, in combination with water, can accelerate and improve the deactivation procedure. The reason for such is that the supplement of CaCl_2_ leads to a faster voltage decrease in comparison with the TW and DW, respectively. After 5 h of insertion time, the CaCl_2_ solution led to a voltage close to 0 V. Insertion in TW resulted in the most noteworthy change between 48 h and the 1st week, where the destructive nail test shows a reduction in temperature, and thus reduced residual energy content, meaning a safe deactivation is possible between 48 h and the 1st week.

Looking at the third deactivation fluid (DW), the inserted cells retained their voltage (close to Umax) for up to 2 weeks. It was only for the last reading at 10.5 weeks where the voltage dropped to 0.62 V for one of the two samples. From the destructive test of these two samples, it was concluded that at least some energy was lost before (up to the 2 weeks interval), with one cell staying below 80 °C and without fire. This means that even though a high voltage was present at 2 weeks, a reduced thermal reaction was observed, leading to the suspicion that the voltage may not be a completely reliable indicator for all situations concerning deactivation.

Accordingly, another reason why the voltage may not be the only parameter is the dynamic reaction of the outer cell terminals with the fluid, especially for CaCl_2_. Here, the risk emerges that a false deduction can be made, as the measurement may deliver a faulty voltage due to contact loss between the terminal and the electrode, and due to incrustations at the terminals. The dissolution of the positive terminal is confirmed by the weight measurements for CaCl_2_. This risk is mainly expressed for CaCl_2_ solution deactivated cells as the other two references (TW and DW cells) did not display such anomalies. Regarding the discharge process, CaCl_2_ solution deactivated cells were initially expected to electrically discharge much faster because of the high volume of the ions in the solution, whereas for TW, this process takes much longer, and for DW, almost no electrical discharge was noticed. Deriving from the results, it seems that for the first two instances, this effect occurs, whereas for DW, not much discharge is expected, and the authors expect the cell to have opened in the period between 48 h and 2 weeks.

When considering the emergence of HF, it is difficult to give a clear statement, as the only fluoride reading above the detection range was measured for TW after 10.5 weeks with 11.3 ppm. For the CaCl_2_ solution deactivated cell, no fluoride was measured because it was captured completely by the calcium ions, which confirms the positive effect of CaCl_2_.

Altogether, the parameters within the circumstances from this investigation can be placed in the following order from most suitable to least suitable: Abuse testing > Voltage > Lithium concentration > Mass > Conductivity > Fluoride concentration > pH. Out of all three solutions, the CaCl_2_ solution yielded the best results in terms of required time for residual energy loss (with 2 h) and cell opening. Additionally, as no F- ions were detected, the authors suppose that if any HF was released, it was likely captured by the calcium ions.

## Figures and Tables

**Figure 1 sensors-23-03901-f001:**
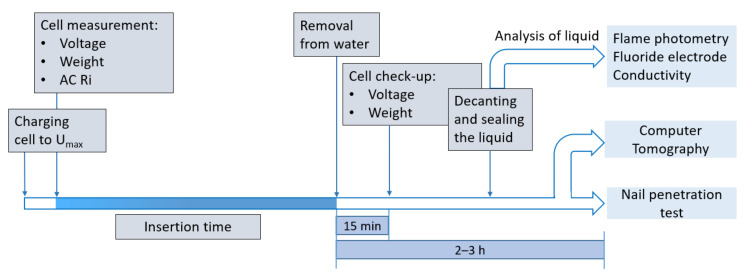
Schematic of the test procedure of each cell.

**Figure 2 sensors-23-03901-f002:**
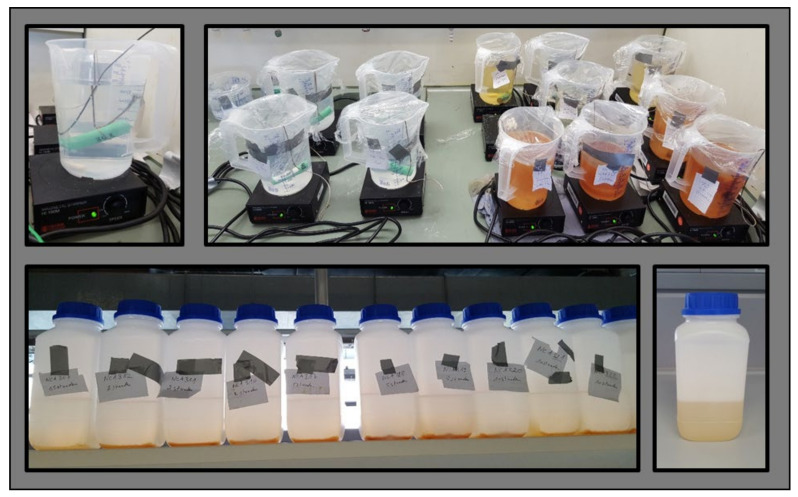
Insertion setup of an 18,650 cell with temperature sensor, fixation of the cell with a string, and constant stirring of the liquid (**top left**); Overview of simultaneous tests, the colours of the liquids indicate either the different duration of tests or different insertion media (**top right**); The collection of filtered liquid in Polyethylene bottles with the lid after the experiment; the components of the liquid already precipitated to the bottom (**bottom left**); Filtered and shaken liquid in Polyethylene bottle prepared for chemical analysis (**bottom right**).

**Figure 3 sensors-23-03901-f003:**
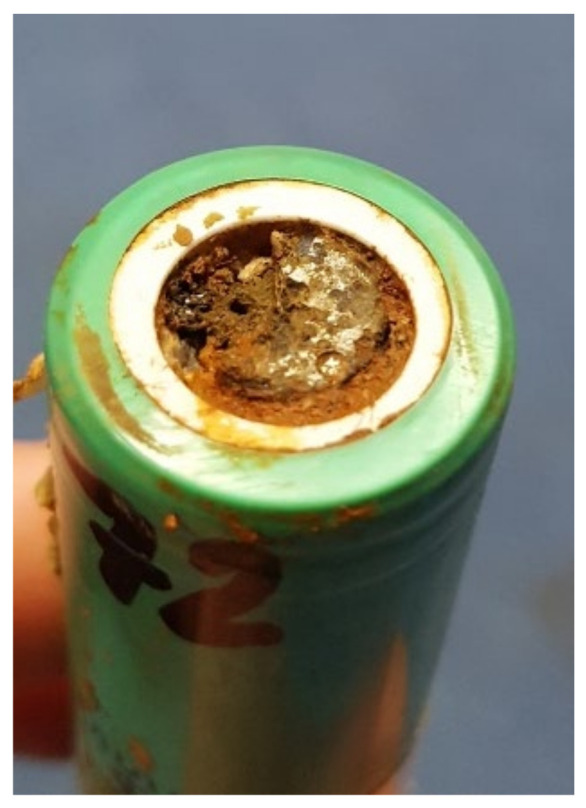
Incrustations at the positive terminal of the cell for TW for two weeks.

**Figure 4 sensors-23-03901-f004:**
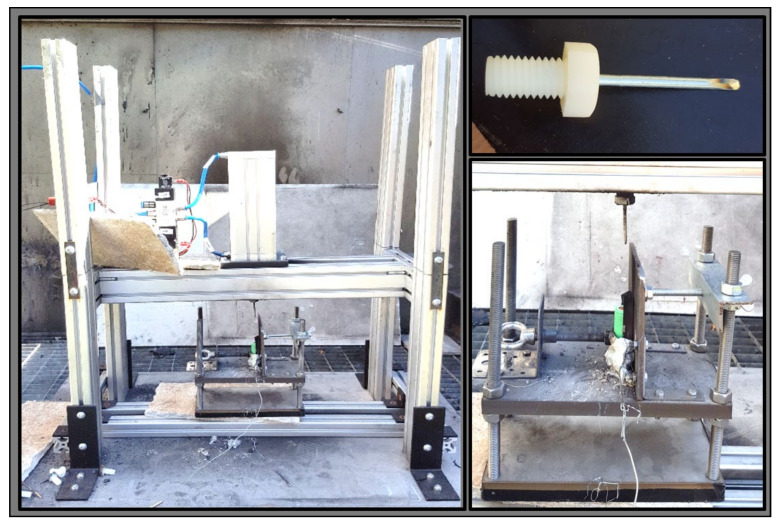
Nail apparatus (**left**); Arrow formed nail (**top right**); Detailed view of the fixation of the cell with rod temperature sensor fixed with plasticize in the middle of the cell (**bottom right**).

**Figure 5 sensors-23-03901-f005:**
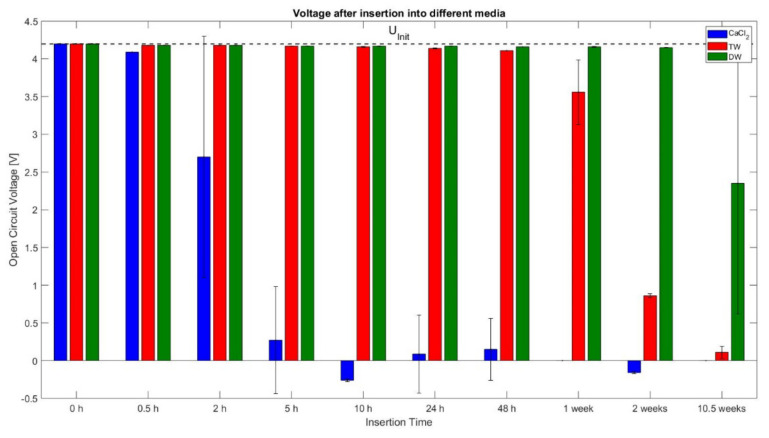
Averages of measured open circuit voltages after insertion into different media (CaCl_2_, TW and DW) with one σ standard deviation; the initial voltage value of each cell U_Init_ of 4.18 V.

**Figure 6 sensors-23-03901-f006:**
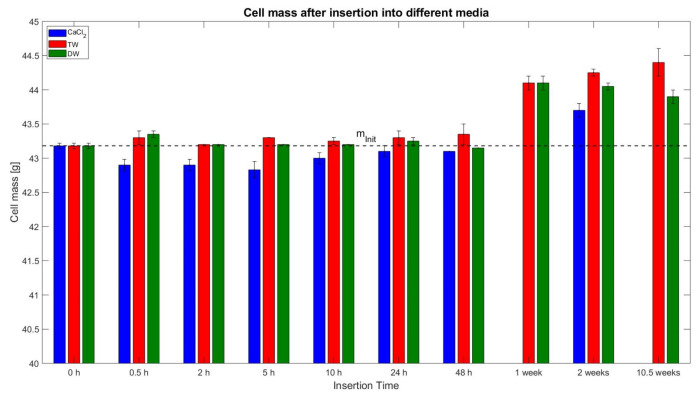
Averages of measured lithium-ion cell mass after insertion into different media (CaCl_2_, TW, and DW) with one σ standard deviation; the initial mass value of each cell m_Init_ of 43.18 g.

**Figure 7 sensors-23-03901-f007:**
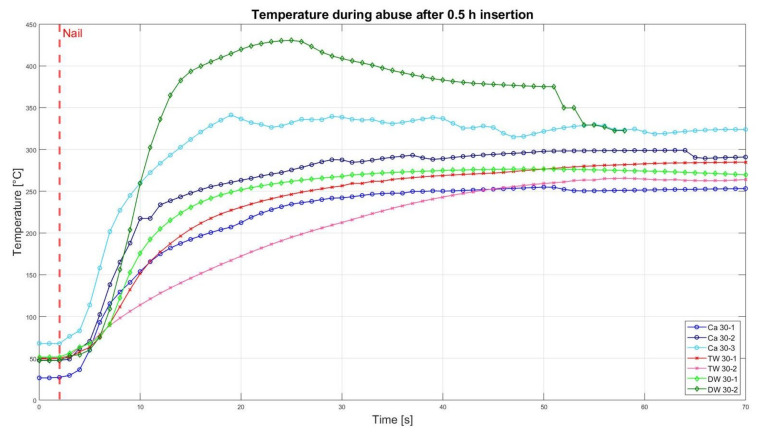
Temperatures during nail penetration test for all cells of category 0.5 h insertion in different insertion media: CaCl_2_ (blue), TW (red), and DW (green), maximum temperature difference: Ca 30-1 = 383.2 °C; Ca 30-2 = 261.9; Ca 30-3 = 273.6 °C; TW 30-1 = 233 °C; TW 30-2 = 213.8 °C; DW-30-1 = 225 °C; DW 30-2 = 383.2 °C.

**Figure 8 sensors-23-03901-f008:**
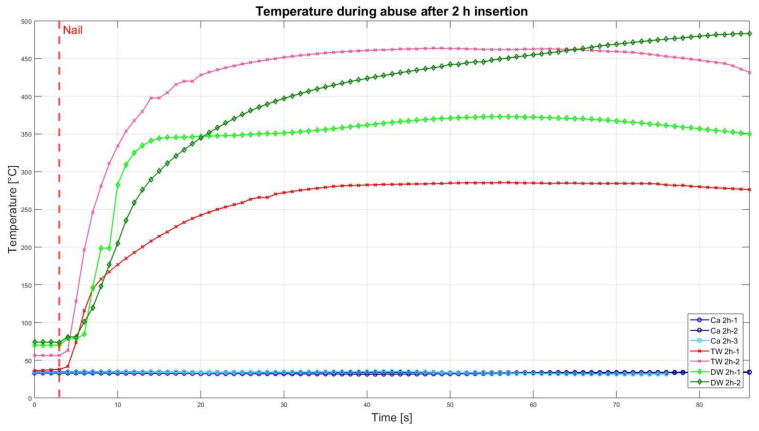
Temperatures during nail penetration test for all cells of category 2 h insertion in different insertion media: CaCl_2_ (blue), TW (red) and DW (green), maximum temperature difference: Ca 2h-1 = 0.9 °C; Ca 2h-2 = 0.1 °C; Ca 2h-3 = 0.3 °C; TW 2h-1 = 247.8 °C; TW 2h-2 = 407.6 °C; DW 2h-1 = 302.9 °C; DW 2h-2 = 402.3 °C.

**Figure 9 sensors-23-03901-f009:**
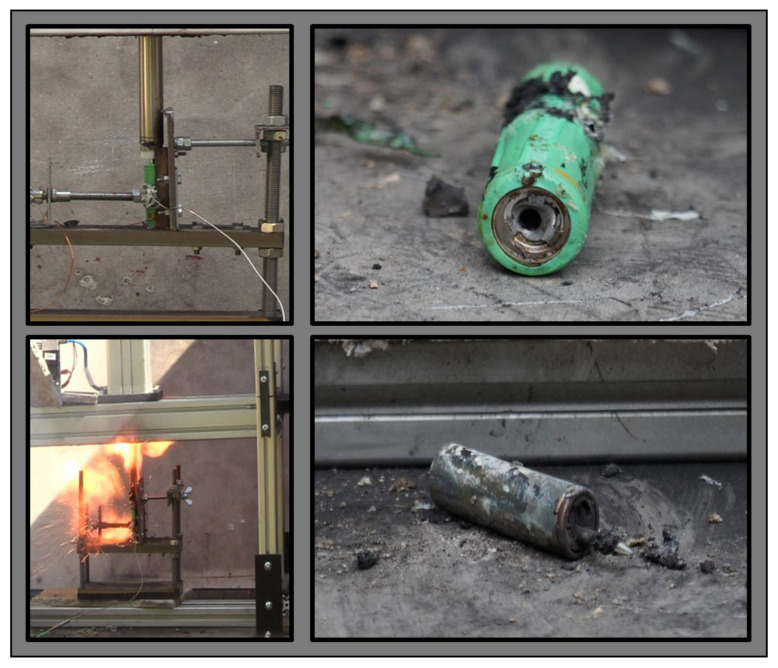
A in CaCl_2_ deactivated cell shows no reaction when the nail penetrates the cell (**top left**); Nail punctured the cell in CaCl_2_ solution in the middle of the positive terminal (**top right**); In DW deactivated cell shows violent reactions when the nail hits the cell (**bottom left**); In DW deactivated cell, which burnt out, nail got stuck in the cell after being pulled out (**bottom right**).

**Figure 10 sensors-23-03901-f010:**
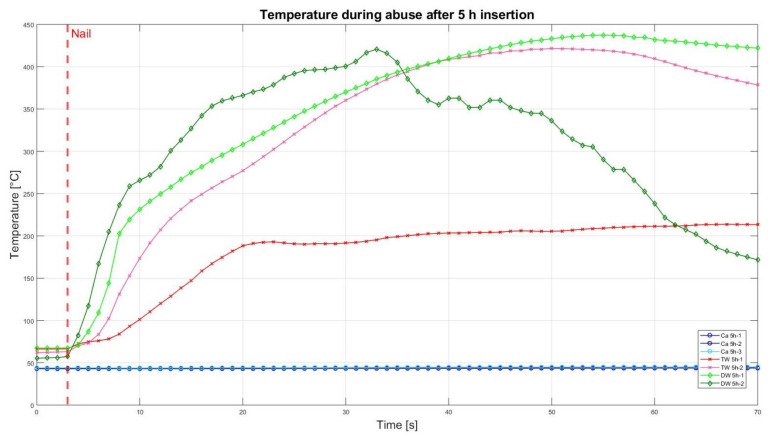
Temperatures during nail penetration test for all cells of category 5 h insertion in different insertion media: CaCl_2_ (blue), TW (red) and DW (green), maximum temperature difference: Ca 5h-1 = 0.7 °C; Ca 5h-2 = 0.5 °C; Ca 5h-3 = 0.6 °C; TW 5h-1 = 150.2 °C; TW 5h-2 = 358.8 °C; DW 5h-1 = 370 °C; DW 5h-2 = 363.3 °C.

**Figure 11 sensors-23-03901-f011:**
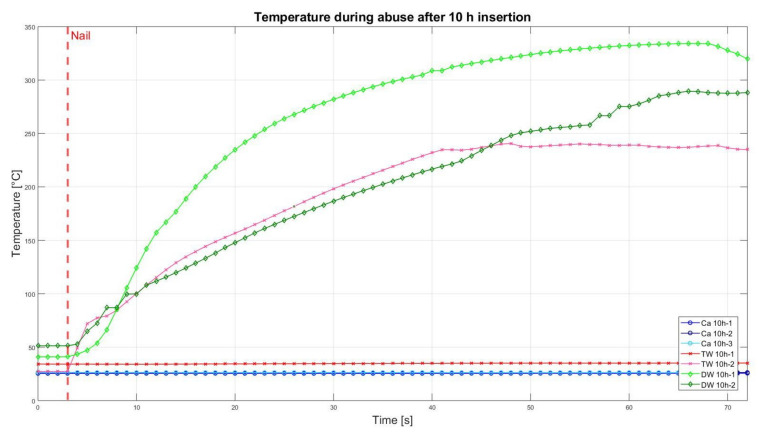
Temperatures during nail penetration test for all cells of category 10 h insertion in different insertion media: CaCl_2_ (blue), TW (red) and DW (green), maximum temperature difference: Ca 10h-1 = 0.1 °C; Ca 10h-2 = 1.4 °C; Ca 10h-3 = 0.2 °C; TW 10h-1 = 0.3 °C; TW 10h-2 = 213.5 °C; DW 10h-1 = 293 °C; DW 10h-2 = 238.3 °C.

**Figure 12 sensors-23-03901-f012:**
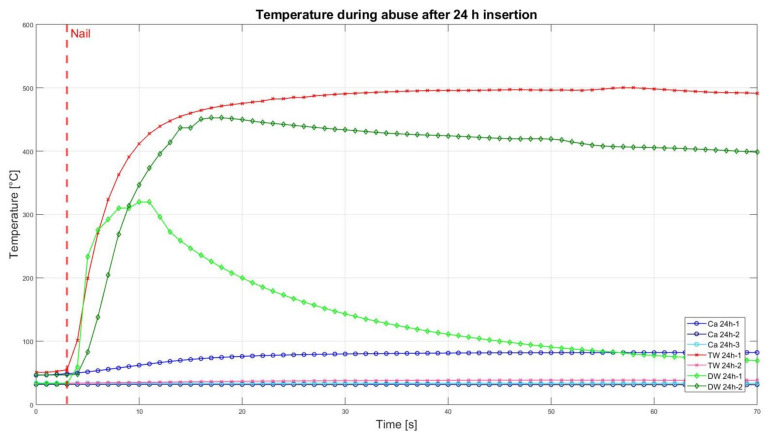
Temperatures during nail penetration test for all cells of category 24 h insertion in different insertion media: CaCl_2_ (blue), TW (red) and DW (green), maximum temperature difference: Ca 24h-1 = 33.4 °C; Ca 24h-2 = 0.1 °C; Ca 24h-3 = 0.4 °C; TW 24h-1 = 445 °C; TW 24h-2 = 4.1 °C; DW 24h-1 = 287.6 °C; DW 24h-2 = 405.8 °C.

**Figure 13 sensors-23-03901-f013:**
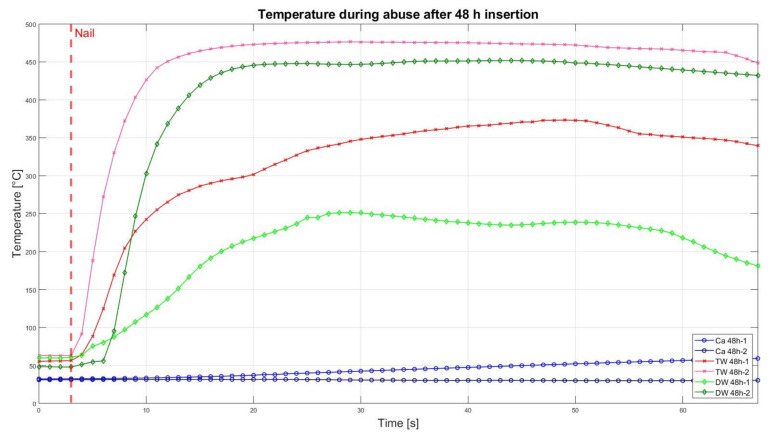
Temperatures during nail penetration test for all cells of category 48 h insertion in different insertion media: CaCl_2_ (blue), TW (red), and DW (green), maximum temperature difference: Ca 48h-1 = 25.5 °C; Ca 48h-2 = 0.2 °C; TW 48h-1 = 317 °C; TW 48h-2 = 413.6 °C; DW 48h-1 = 191.7 °C; DW 48h-2 = 404 °C.

**Figure 14 sensors-23-03901-f014:**
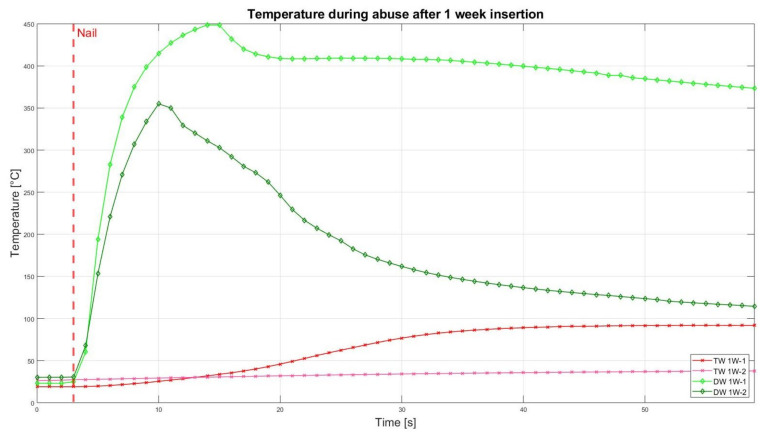
Temperatures during nail penetration test for all cells of category 1 week insertion in different insertion media: TW (red) and DW (green), maximum temperature difference: TW 1W-1 = 72.8 °C; TW 1W-2 = 324.6 °C; DW 1W-1 = 425.5 °C; DW 1W-2 = 0.3 °C.

**Figure 15 sensors-23-03901-f015:**
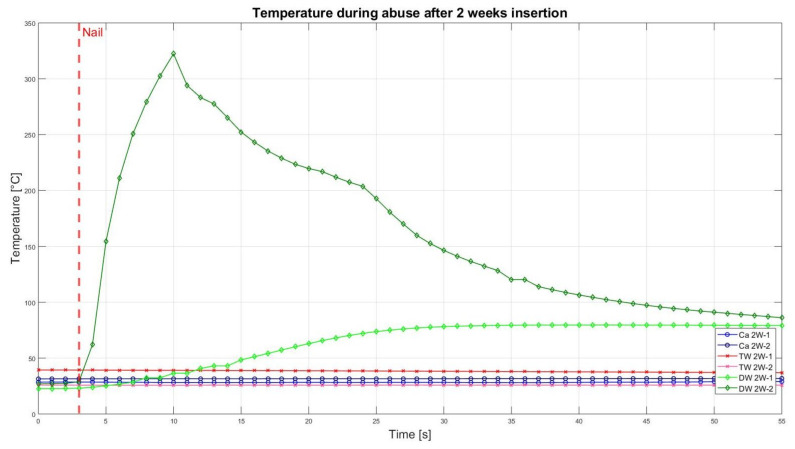
Temperatures during nail penetration test for all cells of category 2 weeks insertion in different insertion media: CaCl_2_ (blue), TW (red), and DW (green), maximum temperature difference: Ca 2W-1 = 0.2 °C; Ca 2W-2 = 293.6 °C; TW 2W-1 = 56.5 °C; TW 2W-2 = 0 °C; DW 2W-1 = −0.5 °C; DW 2W-2 = 0.2 °C.

**Figure 16 sensors-23-03901-f016:**
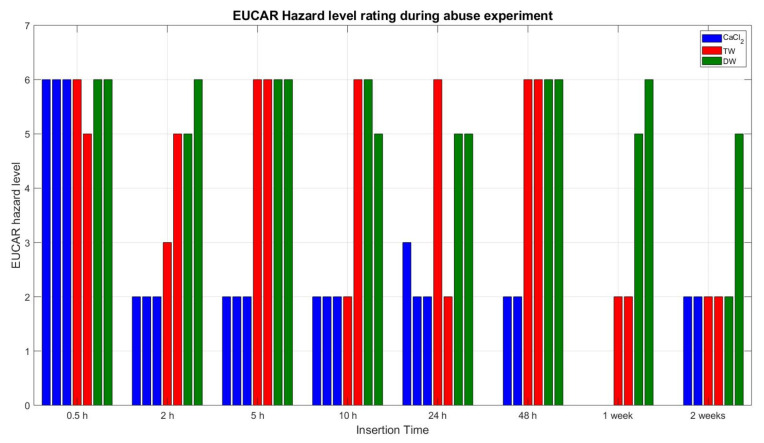
EUCAR hazard rating of all abusive nail penetration experiments, displayed for every single cell; the different colours depict the different deactivation media, blue = CaCl_2_, red = TW, green = DW.

**Figure 17 sensors-23-03901-f017:**
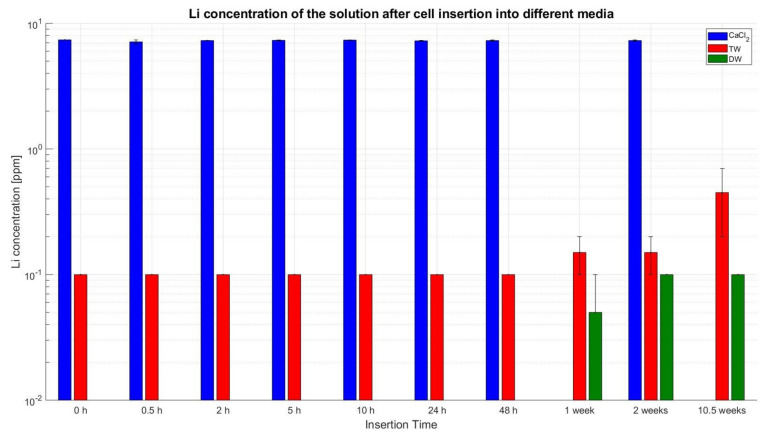
Lithium concentration of the liquid after cell deactivation, cell removal and liquid filtration logarithmically represented; CaCl_2_ (blue), TW (red), and DW (green).

**Figure 18 sensors-23-03901-f018:**
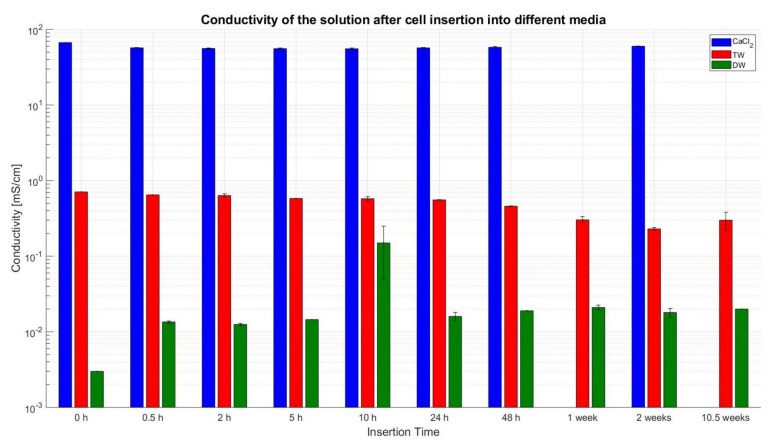
Conductivity of the liquid after cell deactivation, cell removal and liquid filtration; CaCl_2_ (blue), TW (red), and DW (green).

**Figure 19 sensors-23-03901-f019:**
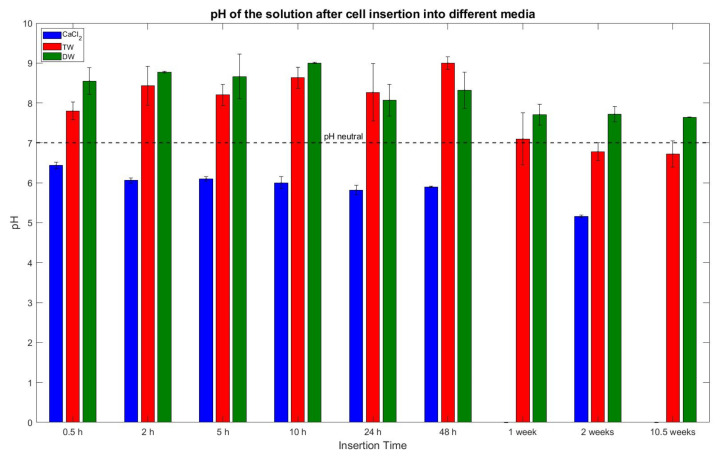
Measured pH values of the liquids with standard deviation. The dashed line indicates the neutral pH value of 7.

**Figure 20 sensors-23-03901-f020:**
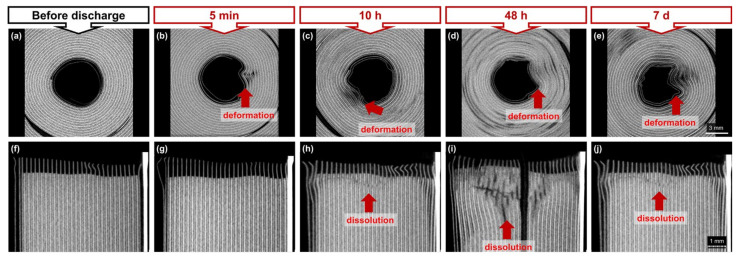
CT cross-sections in (**a**–**e**) x/y-plane and (**f**–**j**) x/z-plane; (**a**,**f**) jelly roll in a pristine state before discharge in CaCl_2_ solution; (**b**–**e**) light to severe deformation of the jelly roll due to storage in CaCl_2_ solution; (**g**) no dissolution of the active material for 5 min storage; (**h**–**j**) light to the severe dissolution of the active material for 10 h, 48 h, and 1 week storage time in CaCl_2_; the deformation of the jelly facilitates the penetration of the CaCl_2_ and leads to the faster dissolution of the active material.

**Figure 21 sensors-23-03901-f021:**
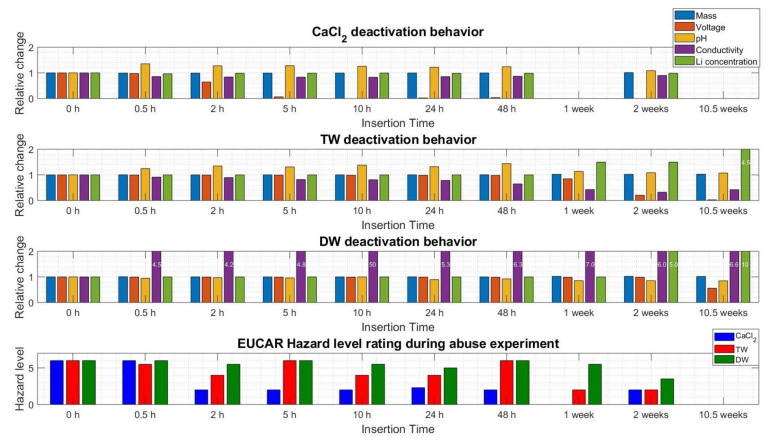
Overview of the relative change of the various parameters recorded during the experiments with all follow-up values displayed in relation to the initial values; the last row displays the change in the EUCAR hazard-rating scheme in absolute numbers; all values depict the average for that condition.

**Table 1 sensors-23-03901-t001:** Overview of the used lithium-ion cells [8].

Type	Nominal Capacity/Ah	Maximum Cut-off Voltage (Umax)/V	Minimum Cut-off Voltage (Umin)/V	Negative Electrode	Manufacturer
LiNiCoAlO_2_ (NCA)	2.5	4.20 ± 0.05	2.5	Graphite	Samsung

**Table 2 sensors-23-03901-t002:** Overview of categories of different Insertion media.

		CaCl_2_	TW	DW
Insertion Time	Total Amount of Liquid	Quantity of Cells	Quantity of Cells	Quantity of Cells
0.50 h	0.4 dm^3^	3	2	2
2.00 h	0.4 dm^3^	3	2	2
5.00 h	0.4 dm^3^	3	2	2
10.0 h	0.4 dm^3^	3	2	2
24.0 h	0.8 dm^3^	3	2	2
48.0 h	0.8 dm^3^	2	2	2
3 weeks	0.8 dm^3^	2	2	2

**Table 3 sensors-23-03901-t003:** Composition of contents in used TW [13].

Components	Amount
Calcium	91 mg/dm^3^
Magnesium	32 mg/dm^3^
Sodium	6.0 mg/dm^3^
Potassium	1.8 mg/dm^3^
Chloride	7.4 mg/dm^3^
Sulphate	30 mg/dm^3^
Nitrate	0.84 mg/dm^3^
Hydrogen carbonate	406.6 mg/dm^3^
Fluoride	0.09 mg/dm^3^
Lead	<0.001 mg/dm^3^

**Table 4 sensors-23-03901-t004:** Sample distribution matrix for all deactivation intervals and various analysis methods, with the symbols representing each deactivation fluid: o for CaCl_2_, x for TW, and ◊ for DW.

Analyses Method/Time Interval	0.5 h	2 h	5 h	10 h	24 h	48 h	1 Week	2 Weeks	10.5 Weeks
Cell voltage	o x ◊	o x ◊	o x ◊	o x ◊	o x ◊	o x ◊	x ◊	o x ◊	x ◊
Cell mass	o x ◊	o x ◊	o x ◊	o x ◊	o x ◊	o x ◊	x ◊	o x ◊	x ◊
Nail penetration	o x ◊	o x ◊	o x ◊	o x ◊	o x ◊	o x ◊	x ◊	o x ◊	
Conductivity	o x ◊	o x ◊	o x ◊	o x ◊	o x ◊	o x ◊	x ◊	o x ◊	x ◊
pH value	o x ◊	o x ◊	o x ◊	o x ◊	o x ◊	o x ◊	x ◊	o x ◊	x ◊
Flame photometer	o x ◊	o x ◊	o x ◊	o x ◊	o x ◊	o x ◊	x ◊	o x ◊	x ◊
Fluoride electrode	o x ◊	o x ◊	o x ◊	o x ◊	o x ◊	o x ◊	x ◊	o x ◊	x ◊
Computer Tomography				o		o	o		

**Table 5 sensors-23-03901-t005:** Measured values and average of the fluoride content of the probes with a minimum resolution range of 0.1 ppm fluoride content, values below 0.1 ppm detection range have no significance [17].

Insertion Time	Deactivation Medium	Measured Fluoride Contents [ppm]	Average Fluoride Content [ppm]
0.5 h	CaCl_2_	0.050	0.033
		0.030	
		0.020	
	TW	0.025	0.024
		0.023	
	DW	0.012	0.014
		0.016	
2 h	CaCl_2_	0.040	0.030
		0.030	
		0.020	
	TW	0.024	0.025
		0.026	
	DW	0.013	0.014
		0.014	
5 h	CaCl_2_	0.050	0.080
		0.090	
		0.100	
	TW	0.021	0.025
		0.028	
	DW	0.012	0.013
		0.014	
10 h	CaCl_2_	0.100	0.113
		0.080	
		0.160	
	TW	0.019	0.019
		0.018	
	DW	0.013	0.025
		0.036	
24 h	CaCl_2_	0.040	0.077
		0.060	
		0.130	
	TW	0.030	0.027
		0.024	
	TW	0.014	0.013
		0.011	
48 h	CaCl_2_	0.130	0.115
		0.100	
	TW	0.029	0.024
		0.019	
	DW	0.016	0.013
		0.010	
1 week	TW	0.046	0.105
		0.163	
	DW	0.030	0.050
		0.069	
2 weeks	CaCl_2_	0.100	0.095
		0.090	
	TW	0.026	0.056
		0.089	
	DW	0.018	0.048
		0.077	
10.5 weeks	TW	12.95	11.306
		9.672	
	DW	0.040	0.043
		0.046	

## Data Availability

Not applicable.

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
