# Peer review of "Analysis of Deactivation of 18,650 Lithium-Ion Cells in CaCl2, Tap Water and Demineralized Water for Different Insertion Times"

_sensors, 2023, doi:10.3390/s23083901_

Round 1

Reviewer 1 Report

In the introduction part, the authors are suggested to supplement the background of using CaCl2 as a salt medium.

The authors should pay attention to the format of the entire manuscript, such as subscripts (line 100, page 3; line 169, page 4, etc.)

The different trend of DW 30-2 for the measured temperatures for 0.5 h deactivated cells (Figure 7 and Figure 10) should be explained.

In Figure 18, the conductivity of CaCl2 solution after insertion time of 10.5 weeks was missing.

The authors declared that no F- ions were detected, and they concluded that released HF was captured by the calcium ions. There is no evidence to support such a conclusion.

Author Response

Many thanks to the reviewer for the great positive comments. We have taken into account all the comments and have made relevant changes to the manuscript. Please see the responses in the uploaded file, 'Author response to reviewer 1- MDPI'.  

Reviewer 2 Report

Overall this is a well written article and the experimental results are clearly explained. One potentially important omission is the lack of fluoride ion concentration for the CaCl2 solution after 10.5 weeks. Measurements from the CaCl2 solutions up to 2 weeks show average fluoride concentrations below the detectable limit (<0.1 ppm) which is also true for the tap water and distilled water solutions, while at 10.5 weeks the average fluoride content for the tap-water solution is much higher (11.3 ppm) than any previous measurement, suggesting electrolyte leakage into the solution. The utility of CaCl2 as a method to remove HF from the deactivation is a key motivation of the study, but without the 10.5 week comparison this motivation is not fully evaluated. Alternatively, deliberately spiking tap water or distilled water with a known concentration of LIB electrolyte and measuring fluoride concentration with and without CaCl2 present would be another method to evaluate the efficacy of the CaCl2 salt. 

Author Response

Many thanks to the reviewer for the great positive comments. We have taken into account all the comments and have made relevant changes to the manuscript. Please see the responses in the uploaded file, 'Author response to reviewer 2- MDPI'.  
